# Neural Circuitry–Neurogenesis Coupling Model of Depression

**DOI:** 10.3390/ijms22052468

**Published:** 2021-02-28

**Authors:** Il Bin Kim, Seon-Cheol Park

**Affiliations:** 1Department of Psychiatry, Hanyang University Guri Hospital, Guri 11923, Korea; jonecaby49@gmail.com; 2Graduate School of Medical Science and Engineering, Korea Advanced Institute of Science and Technology, Daejeon 34141, Korea

**Keywords:** depression, entorhinal cortex, hippocampus, memory, pattern separation, mood, neural circuitry, neurogenesis

## Abstract

Depression is characterized by the disruption of both neural circuitry and neurogenesis. Defects in hippocampal activity and volume, indicative of reduced neurogenesis, are associated with depression-related behaviors in both humans and animals. Neurogenesis in adulthood is considered an activity-dependent process; therefore, hippocampal neurogenesis defects in depression can be a result of defective neural circuitry activity. However, the mechanistic understanding of how defective neural circuitry can induce neurogenesis defects in depression remains unclear. This review highlights the current findings supporting the neural circuitry-regulated neurogenesis, especially focusing on hippocampal neurogenesis regulated by the entorhinal cortex, with regard to memory, pattern separation, and mood. Taken together, these findings may pave the way for future progress in neural circuitry–neurogenesis coupling studies of depression.

## 1. Introduction

Depression is the leading cause, only second to heart diseases, of disability worldwide [1]. The global prevalence of depression is 4.4%, and the total estimated number of patients with depression increased by 18.4% between 2005 and 2015 [2,3]. Although effective treatments have been available for some patients, the prevalence of depression has remained remarkably stable for decades, with low remission and high relapse rates [4,5,6,7]. The therapeutic challenge in current psychiatry has spurred a concept of next-generation treatment that emphasizes more refined and individualized clinical treatments based on biological markers and endophenotypes, rather than just categorical diagnoses [8]. Correspondingly, the current therapeutic outcomes require further understanding of depression pathophysiology to facilitate theoretical paradigms that apply to refractory or partially remitted depression. For decades, there have been leading paradigms, including monoamine chemistry, neural plasticity, and neural circuitry, each of which paves the way for not just theoretical but also clinically applicable targets, such as neurotransmitters, neural stem cell niches, and specific brain regions and networks, all crucial to depression-related behaviors in both humans and animals. The contemporary paradigms can progress via an effort to understand depression pathophysiology as a joint model between neurotransmitters, stem cell niches, and brain networks. Therefore, the neural circuitry-regulated neurogenesis via synaptic neurotransmission can be a pioneering theme to enhance our understanding of depression pathophysiology.

Hippocampal neurogenesis defects are a hallmark of depression. Mounting evidence points to hippocampal deficits in activity and volume and reductions in activity-dependent gene expression, which collectively reflect neurogenesis defects in the dentate gyrus of the hippocampus [9,10,11]. Hippocampal neurogenesis conceives neural circuitry activity-mediated regulation [12,13,14]; therefore, there have been exploratory efforts to stimulate specific upstream hippocampal circuitries in order to enhance neurogenesis in patients with depression as well as stressed animals. Intriguingly, in animal models, approaches such as deep brain stimulation demonstrate that stimulation of the entorhinal cortex enhances hippocampal-dependent cognitive performance through hippocampal neurogenesis, resulting in improvements in memory and pattern separation [15,16,17]. These findings in animal models corroborated the entorhinal cortex-regulated hippocampal neurogenesis and memory enhancement in human subjects [18]. Furthermore, pioneering studies using advanced techniques, including optogenetics, chemogenetics, and molecular-based approaches, have highlighted the entorhinal–hippocampal circuitry in the regulation of neurogenesis and hippocampal-dependent cognitive and emotional functions. For a representative example, precise stimulation of entorhinal glutamatergic afferents leads to improvements in depression-related behaviors in stress-inoculated animal models, which is accompanied by increased hippocampal neurogenesis [19]. Taken together, entorhinal cortex–hippocampal circuitry implicates neurogenesis in memory, pattern separation, and mood, all of which can be hampered in depression. Thus, the entorhinal–hippocampal circuitry may be a plausible model of neural circuitry–neurogenesis in depression. This review highlights findings from animal and human studies, to support the causal relationship between entorhinal–hippocampal circuitry and neurogenesis in the regulation of memory, pattern separation, and mood. Then, suggest future directions of the neural circuitry–neurogenesis coupling model of depression.

## 2. Activity-Dependent Adult Neurogenesis and Framework of Neural Circuitry–Neurogenesis Coupling Model of Depression

Our formulation of the neural circuitry–neurogenesis coupling model of depression is based on both long-lasting neurophysiological knowledge of adult hippocampal neurogenesis and pioneering works not just recapitulating neurogenesis but also demonstrating the impact of neural circuitry modification on neurogenesis and depression-related behaviors. Here, we review the neurophysiological findings supporting that the entorhinal cortex and hippocampus collectively contribute to adult neurogenesis. The entorhinal cortex and hippocampus are synaptically involved in adult neurogenesis [20,21,22,23], implicating a long-range network in the regulation of neural stem cell niches. Adult neurogenesis is characterized by a dynamic capacity to modify the synaptic strength and number, which is regulated by diverse mechanisms that render synaptic inputs in an activity-dependent manner to the hippocampus [12,14,24]. The entorhinal cortex provides the major excitatory input to the dentate gyrus, the only hippocampal subregion, where granule cells are newly generated, become mature, and are finally incorporated into existing hippocampal circuitry. Glutamatergic stimulation of the hippocampus has long been implicated in the regulation of adult neurogenesis at multiple stages [25]. Early work using a patch-clamp recording demonstrated robust glutamatergic synaptic connectivity between granule cells and entorhinal projections, which occurs at 2–3 weeks of neuron age after the stimulation of the entorhinal projection to the hippocampal granule cells [26]. This is in line with another finding from a trans-synaptic tracing study [27]. Intriguingly, the entorhinal cortex only begins to develop synaptic input to adult-born neurons when the adult-born neurons reach 21 to 28 days of cell age, during which maturation of the neurons occurs [27]. This critical period suggests that the entorhinal cortex is implicated in the regulation of adult neurogenesis during the maturation phase. This is also supported by a study that identified this critical period during which glutamatergic stimulation of perforant paths linking the entorhinal cortex to the hippocampal dentate gyrus, which leads to enhanced long-term potentiation of adult-born neurons [28,29]. During the 4–6 weeks after birth, neurons exhibit both a lower threshold and a higher long-term potentiation amplitude by physiological levels of stimulation. Even though there might be indirect regulations through a non-cell-autonomous mechanism that modulates existing neural circuitry, these are beyond the scope of this review. Altogether, the entorhinal cortex has a regulatory role in hippocampal adult neurogenesis in an activity-dependent manner (Figure 1). Thus, the concept of entorhinal–hippocampal circuitry and subsequent neurogenesis is tenable. Based on this concept, we address the question of whether the neural circuitry–neurogenesis coupling model can apply to depression pathophysiology, which might be in part supported by antecedent works pointing to hippocampal neurogenesis defects as a hallmark of depression [30,31,32]. To support this idea, we will review current works with a focus on entorhinal–hippocampal circuitry, neurogenesis, and depression-related phenotypes, including memory, pattern separation, and mood.

## 3. Entorhinal–Hippocampal Circuitry and Neurogenesis in Memory

Episodic memory damage is a major cognitive symptom of depression [33,34,35,36,37,38,39,40]. Episodic memory deficits relate to volume reductions not just in the hippocampus [41,42,43,44], but in the entorhinal cortex [45], indicating that cognitive symptoms of depression might be partially derived from combined pathophysiology encompassing the entorhinal cortex and hippocampus. In particular, numerous studies have suggested that hippocampal neurogenesis defects lead to impaired episodic memory in depression [46,47,48,49]. Nonetheless, the mechanism by which upstream hippocampal circuitry regulates episodic memory through hippocampal neurogenesis remains unclear.

### 3.1. Supportive Findings from Human Studies

Entorhinal–hippocampal circuitry is known as a memory hub of human and primate brains, mainly in processing episodic memories of objective, spatial, and temporal information, such as what, where, and when [50,51,52,53,54,55]. In humans, it has been demonstrated that stimulation of the entorhinal cortex prompts favorable physiological changes, including memory- and learning-related processes. Specifically, deep brain stimulation of the human entorhinal cortex induces enhancements in spatial memory [18]. In a spatial navigation task, human subjects with entorhinal stimulation reached a destination within a virtual environment in a shorter time compared with controls without entorhinal stimulation (Figure 2a). Notably, entorhinal stimulation is accompanied by the resetting of the hippocampal theta rhythm which allows optimal conditions for the induction of long-term potentiation, giving rise to fine hippocampal encoding of spatial information [56]. In contrast, direct deep brain stimulation in the hippocampus does not affect or impair hippocampus-dependent memory processing [18,57], thus emphasizing the efficacy of targeting upstream hippocampal circuitry rather than the hippocampus per se.

### 3.2. Supportive Findings from Animal Studies

In addition to this finding in humans, mouse studies also demonstrated that deep brain stimulation of the entorhinal cortex leads to improvements in spatial learning and memory that are accompanied by enhanced neurogenesis in the dentate gyrus (Figure 2b). Specifically, Stone and colleagues reported that transient deep brain stimulation of the entorhinal cortex with high frequencies activates the neural stem cell niche to yield sequential neurogenesis processes, including proliferation of the dentate gyrus, progeny cell differentiation into neurons, survival of the neurons for at least several (>5) weeks, and the maturation of neurons into dentate granule cells [15]. Importantly, the dentate granule cells are finally, but in a delayed manner, integrated into the hippocampal circuitry after stimulation of the entorhinal cortex. Correspondingly, in the Morris water navigation task, spatial memory implicated in the hippocampal circuitry is established six weeks rather than one week after the entorhinal stimulation. This delayed effect of the entorhinal stimulation matches the maturation-dependent integration of adult-born granule cells into the hippocampal circuitry, thereby supporting spatial memory [58,59]. Researchers finally emphasized a causal relationship between entorhinal stimulation-dependent hippocampal neurogenesis and enhanced spatial memory by attempting to block neurogenesis prior to assessing whether spatial memory is enhanced or not. The impact of the entorhinal stimulation on adult hippocampal neurogenesis and spatial memory is also supported by studies that used a similar approach [17,18].

In addition to the deep brain stimulation approach, preclinical research has employed an optogenetic approach to gain further insight into the detailed physiology of memory implicated in entorhinal–hippocampal circuitry. Robinson and colleagues questioned whether entorhinal–hippocampal circuitry regulates temporal memory that is encoded in the principal cells, also known as time cells, of the hippocampal CA1 region [60]. They investigated whether optogenetic inactivation of the medial entorhinal cortex results in the disruption of hippocampal CA1 temporal encoding and memory across time (Figure 2c). The medial entorhinal cortex provides a major cortical input to the hippocampus for processing not only space but also time information, in parallel with the lateral entorhinal cortex for object information [61,62,63,64,65]. They implemented bilateral optic fiber arrays for light-delivered silencing of the medial entorhinal cortex while simultaneously recording hippocampal CA1 regions in rats treated with bilaterally targeted adeno-associated viral vector to the medial entorhinal cortex. In a complex behavioral task of sequential object–treadmill–maze phases, they evaluated the impact of medial entorhinal inactivation on hippocampal CA1 coding activity for object, time, and space information in order. Rats were exposed to a specific object for a short period and then sent onto a treadmill to run for an intended time delay prior to the second exposure to the object. Accordingly, temporal memory was evaluated during the treadmill phase in a fixed-space environment. Strikingly, medial entorhinal cortex inactivation provoked disruption only in CA1 time coding activity but not in object and space coding activity. This finding indicates a distinct mechanism of entorhinal–hippocampal circuitry by which temporally structured experiences are organized to be a part of episodic memory. Taken together, current preclinical studies support the idea that entorhinal–hippocampal circuitry is crucial for hippocampal-dependent episodic memory. Nonetheless, our understanding of the circuitry mechanism contributing to memory pathophysiology in particular relation to depression is still in its infancy, thus requiring more research using stressed animal models and optogenetic or chemogenetic approaches to address memory deficits in depression models based on entorhinal–hippocampal circuitry and neurogenesis.

Further research is required to address these questions: which specific cell types, or neurotransmitters, of the entorhinal cortex and hippocampus are implicated in spatial and temporal memory encoding defects in stress animal models, respectively; whether temporal memory encoding in the hippocampus still relies on neurogenesis; how the entorhinal cortex drives specific electrophysiological changes, such as hippocampal theta rhythm oscillations, in association with long-term potentiation of spatiotemporal memory during neurogenesis in stressed animals and how the rhythm changes are associated with memory deficits in the depression models; and ultimately, whether therapeutic targeting of the entorhinal–hippocampal circuitry using brain stimulation or pharmacological approaches recovers the spatiotemporal memory deficits with substantial validity and reliability, specifically in humans with depression.

## 4. Entorhinal–Hippocampal Circuitry and Neurogenesis in Pattern Separation

Pattern separation is the ability to distinguish between similar contextual representations and is dependent on hippocampal dentate gyrus neurogenesis [66,67,68,69,70]. Impaired pattern separation is a potential marker for hippocampal neurogenesis defects in depression [71,72]. Neurogenesis ablation studies and contemporary chemogenetic approaches consistently point to the relationship between neurogenesis defects and deficits in pattern separation. On another level, functional imaging studies in association with behavioral tasks additionally support the idea that the entorhinal cortex may be implicated in pattern separation; this is also supported by neurophysiological knowledge that the entorhinal cortex serves as a key region mediating communication between the hippocampus and neocortex to receive and store multimodal cortical sensory and spatial representations before transmitting it to the hippocampal dentate gyrus, where previous and new incoming representations of similar subjects are distinguished through a sparse, flexible coding for diverse activity patterns of different representations [69,73]. Nonetheless, very few studies have directly investigated the relationship between entorhinal–hippocampal circuitry and pattern separation. In this section, we review representative studies that highlight the imaginal correlates of the entorhinal cortex for pattern separation, the impact of neurogenesis defects on pattern separation, and, lastly, a chemogenetic approach to elucidate a causal relationship between entorhinal–hippocampal circuitry, neurogenesis, and pattern separation.

### 4.1. Supportive Findings from Human Studies

Functional imaging studies raise the possibility that the entorhinal cortex may be implicated in the upper hippocampal circuitry to operate pattern separation. Earlier studies used elderly human subjects to examine the association between the entorhinal cortex-implicated circuit dysfunction and cognitive declines, because volume reductions in medial temporal lobes including entorhinal cortex have been found to relate to cognitive deficits in both depression [74] and aging [75,76]. One study using high-resolution functional magnetic resonance imaging (fMRI) explored the possibility of entorhinal–hippocampal circuitry dysfunction leading to pattern separation defects by examining the functional activities of the anterolateral entorhinal cortex and hippocampal dentate gyrus and CA3 (Figure 3a) [77]. In a discrimination task, the subjects with functional entorhinal–hippocampal dissociation were unable to distinguish between similar object representations, although their spatial distinction was intact. Specifically, the subjects showed hypoactivity in the anterolateral entorhinal cortex and, inversely, hyperactivity in the hippocampal dentate gyrus and CA3, suggesting that regional activity imbalances may be related to object discrimination defects. This is corroborated by lateral and medial entorhinal cortex physiologies implicated in object and spatial discrimination, respectively. This is also consistent with a study that used rodents to demonstrate that lateral entorhinal neuronal activity is directly related to hippocampal CA3 hyperactivity, although the human fMRI study rendered only indirect measures of neuronal activity from blood-oxygen-level-dependent (BOLD) values [78]. Another fMRI study also showed that the entorhinal cortex relates to pattern separation in elderly human subjects with or without a diagnosis of depression, with a focus on the basolateral amygdala, hippocampal dentate gyrus/CA3, and lateral entorhinal cortex (Figure 3a) [79]. Subjects with depression showed hypoactivity in the amygdala and, inversely, hyperactivity in the entorhinal cortex as well as the hippocampus during false discrimination of positive similar representations. The authors suggested that the entorhinal cortex may be involved in the emotional processing of pattern separation, and that there may be upstream circuitry encompassing the amygdala and entorhinal cortex for controlling the hippocampal dentate gyrus/CA3. Together, even though these fMRI experiments using elderly humans only noted functional correlates of pattern separation, the results indirectly provide evidence that the entorhinal cortex may be involved in hippocampal circuitry related to pattern separation. Accordingly, a further approach is required to directly elucidate a causal relationship between the entorhinal–hippocampal circuitry and pattern separation and to comprehensively address whether neurogenesis still mediates between the neural circuit and the cognitive manifestation.

### 4.2. Supportive Findings from Animal Studies

Among the earliest approaches in the study of the impact of neurogenesis defects on pattern separation, hippocampal X-ray irradiation or the disruption of synaptic plasticity in dentate granule cells was mostly adopted to recapitulate neural stem cell niche dysfunction to scrutinize impairment in the discrimination of a safe, similar context from the foot-shock context [72,80,81,82,83]. Specifically, Clelland and colleagues applied hippocampal-directed X-ray irradiation to a neurogenesis-ablated mouse model and demonstrated pattern separation defects using spatial discrimination and maze tasks (Figure 3b) [70]. The neurogenesis-ablated mice demonstrated an impaired ability to detect subtle differences between two similar contexts in both tasks. This is in line with independent studies in which hippocampal X-ray-irradiated mice demonstrated an impaired ability of pattern separation in contextual fear conditioning tasks [72,83]. The neurogenesis-ablated mice showed similar freezing behavior between a shock-associated context and a similar no-shock context compared with controls that were able to discriminate between the two contexts. These consistent findings spurred the question that enhancing neurogenesis may increase pattern separation. Accordingly, Sahay and colleagues developed transgenic mice to selectively enhance adult neurogenesis [72]. In a contextual fear conditioning task, the transgenic mice with functionally integrated adult-born dentate neurons showed significantly enhanced performance in discriminating between similar contexts. Taken together, the neurogenesis ablation and genetic modification of neurogenesis function approaches support the idea that pattern separation is dependent on hippocampal neurogenesis. Nevertheless, the understanding of upper hippocampal circuitry to achieve pattern separation remains unclear.

Yun and colleagues adopted a chemogenetic approach to develop a transgenic mouse model with entorhinal cortex-specific knockdown of a psychosocial stress-induced protein using adeno-associated virus-mediated gene transfer (Figure 3c) [19]. Among various stress-induced proteins, they strategically employed *Trip8b*, a knockdown of which is known to increase the excitability of hippocampal neurons, enabling dentate gyrus neurogenesis. In the knockdown mice, stimulation of entorhinal glutamatergic afferents led to activity-dependent hippocampal neurogenesis, including both the generation and dendritic maturation of adult-born dentate gyrus neurons through enhanced intrinsic excitability of stellate cells in the entorhinal cortex. Then, a behavioral task using contextual fear conditioning was adopted to examine the knockdown mouse’s ability (pattern separation) to discriminate the foot shock-associated context from a safe, similar context. The knockdown mice (*Trip8b*-shRNA) exhibited approximately 50% more freezing in the foot shock-paired context compared with the control mice (SCR-shRNA), thus revealing enhanced pattern separation from glutamatergic entorhinal stimulation. Furthermore, dentate gyrus-directed image-guided X-ray irradiation, reflecting neurogenesis ablation, significantly blunted the effect induced by entorhinal-specific *Trip8b* knockdown. Together, this pioneering work using a chemogenetic approach supports the idea that entorhinal–hippocampal circuitry regulates neurogenesis, thereby enabling pattern separation. Nonetheless, our understanding of the circuitry-regulated neurogenesis mechanism underlying pattern separation remains rudimentary; thus, this field of preclinical research requires more efforts to exploit brain stimulation and optogenetic and chemogenetic approaches to delineate the mechanistic comprehension of entorhinal–hippocampal circuitry giving rise to pattern separation.

## 5. Entorhinal–Hippocampal Circuitry and Neurogenesis in Mood Regulation

Hippocampal neurogenesis is closely linked to the pathophysiology of depression as well as the response to antidepressant treatments [84,85]. Hippocampal neurogenesis defects are accompanied by depression-related behaviors, such as helplessness and hopelessness [86,87]. Studies indicate that neurogenesis defects mediate alterations of physiology including inflammation [88,89], the hypothalamic–pituitary–adrenal axis [90], and neurotrophic factors [91,92], which are all associated with depression or stress resilience. Additionally, neurogenesis defects are known to lower the therapeutic effects of antidepressants, delaying recovery from depression [93]. There are, accordingly, efforts to enhance hippocampal neurogenesis, for which stimulation approaches are adopted to induce behavioral effects in stressed animal models. In such trials, the hippocampus is subjected to deep brain stimulation to enhance neurogenesis, which, however, does not result in alterations in hippocampal-dependent functions such as memory [18,57]. Rather, upper hippocampal circuitry such as the entorhinal cortex emerged as a more appropriate target of brain stimulation approaches in ameliorating depression-related behaviors; studies indicate that deep brain stimulation of the entorhinal cortex leads to improved hippocampal-dependent memory [18,57]. Therefore, the mechanism by which the upper hippocampal circuitry including the entorhinal cortex regulates dentate gyrus neurogenesis and relates to depressive symptoms needs to be delineated. A pioneering study has shed light on some clues regarding the mechanistic understanding of the causal relationship between entorhinal–hippocampal circuitry, neurogenesis, and depression-related behaviors in animal models [19].

To date, there are no available studies that exploit deep or superficial brain stimulations targeting the entorhinal cortex to conceptually link hippocampal neurogenesis and anti-depressive behaviors. Most studies have focused on other brain regions, including the prefrontal cortex [94,95,96,97,98,99,100,101,102,103,104,105,106,107], cingulate cortex [108,109,110,111,112,113,114,115], nucleus accumbens [116,117,118,119,120,121], thalamus [122,123,124,125,126], and striatum [127,128,129,130,131], in efforts to search for neural circuitry dysfunctions, which greatly contribute to the current understanding of depression as circuitopathy [132]. Considering the concept of circuitopathy, furthering a perspective of hippocampal neurogenesis can at least complement the understanding of depression circuitopathy to elucidate the impact of the neural circuitry on neurogenesis as well as the relationship between neural circuitry–neurogenesis coupling and anti-depressive effects.

### Supportive Findings from Animal Studies

Chemogenetic or optogenetic stimulation with behavioral tasks can be an optimal option to untangle the causal relationship between entorhinal–hippocampal circuitry and neurogenesis and anti-depressive behaviors. Yun and colleagues showed that chemogenetic and molecular-based stimulation of the entorhinal cortex combats depression-related behaviors in animals under acute and chronic stressful conditions (Figure 4) [19]. They designed an experiment considering that modulation of hippocampal neuronal activity may increase dentate gyrus neurogenesis and mature dendritic morphology, and these neural changes may lead to anti-depressive behaviors. The authors employed *Trip8b*, a specific stress-induced protein that affects hippocampal neuron activity. Indeed, *Trip8b* germline knockout mice demonstrated increased hippocampal neuron firing frequency and more neurogenesis and new neuron maturation than controls, particularly in the temporal dentate gyrus, which is a hippocampal subregion associated with emotion processing and response to stress valences [133,134]. These findings indicate that entorhinal cortex-specific *Trip8b* knockdown enhances dentate gyrus neurogenesis in an activity-dependent manner that is modulated by entorhinal cortex afferents to the dentate gyrus. To address the behavioral effects of *Trip8b* knockdown mice, a forced swimming test and novelty-suppressed feeding test were adopted in different stress inoculations, including basal state, acute restraint stress, and chronic stress with long-term exposure to corticosterone [135,136]. Under all stress states, entorhinal cortex-specific *Trip8b* knockdown promoted anti-depressive behaviors, which are presented both by lower immobility in the forced swimming test and shorter latency to feed in the novelty-suppressed feeding test. The entorhinal–hippocampal circuitry was further scrutinized based on a chemogenetic approach to delineate the glutamatergic or non-glutamatergic neurons responsible for the anti-depressive behaviors. They produced Gq-coupled modified human M3 muscarinic receptor-infused mice that engaged in glutamatergic neurotransmission and mCherry-infused mice that served as controls. The authors demonstrated CamKIIα-iCre-driven mCherry expression [137] exclusively in the entorhinal cortex and hippocampal dentate gyrus, indicating the appropriateness of targeting the entorhinal-dentate gyrus circuit in both mice, and also a higher abundance of c-Fos^+^ cells in the entorhinal cortex and dentate gyrus in the modified M3 muscarinic receptor-infused mice relative to the mCherry-infused control mice, indicating an enhancement in glutamatergic neuronal activity that is followed by the designer ligand administration of clozapine-N-oxide. Intriguingly, chronic chemogenetic stimulation of glutamatergic entorhinal afferents to the dentate gyrus promotes anti-depressive-like behaviors under basal and stress conditions. In a novelty-suppressed feeding test, the modified M3 muscarinic receptor-infused mice exhibited an approximately 50% shorter latency to feed compared with mCherry-infused mice after five weeks, rather than three weeks of clozapine-N-oxide treatment. Additionally, the modified M3 muscarinic receptor-infused mice showed more interaction time with a social target compared with the mCherry-infused mice, after both mice were subjected to a situation mimicking the chronic social defeat model of depression [138]. These findings suggest that glutamatergic entorhinal–hippocampal circuitry regulates hippocampal neurogenesis leading to anti-depressive behaviors, even in stress-inoculated animals ethologically equivalent to depression. Taken together, although still in its infancy, the preclinical work exploiting a chemogenetics-based stimulation approach would establish the distinguished field of depression circuitry, specifically involving the entorhinal cortex and hippocampal dentate gyrus that converges into neural circuitry-regulated neurogenesis, which might be a new key to anti-depressive behaviors.

## 6. Suggestions for Advancing the Neural Circuitry–Neurogenesis Coupling Model of Depression

Pioneering preclinical studies linking entorhinal–hippocampal circuitry, adult neurogenesis, and emotional and cognitive symptoms of depression proposed a first step for the neural circuitry–neurogenesis coupling model in depression. Nonetheless, more efforts should be made to elucidate the mechanism of the depression model in two parallel directions. Firstly, future research is required to delineate the entorhinal–hippocampal circuitry in detail, considering the complex neurochemical physiology between the entorhinal cortex and hippocampal dentate gyrus. During adult neurogenesis, the entorhinal cortex provides major glutamatergic afferents to the hippocampal dentate gyrus, through which the progeny cells of the hippocampus mature to become granule cells that can be integrated into hippocampal circuits [20,27,139,140,141]. Meanwhile, some evidence indicates that the entorhinal cortex also provides GABAergic afferents to the hippocampus, which constitutes entorhinal–hippocampal inhibitory circuitry that controls rhythmic theta activity of post-synaptic neurons in the hippocampal dentate gyrus [142,143]. How the excitatory and inhibitory entorhinal afferents can corroborate neurogenesis and hippocampal-dependent anti-depressive behaviors remains elusive, thus demanding further research. Electrophysiologically, entorhinal electric changes, including gamma rhythm oscillations provoked by hippocampal theta rhythm alterations, modulate long-term potentiation for hippocampal-dependent cognitive functions, specifically, memory and learning [144]. How the electrical rhythm changes across the entorhinal cortex and hippocampus modulate neurogenesis and hippocampal-dependent emotional behaviors remains unresolved. Anatomically, the entorhinal cortex has lateral and medial subdivisions that are known to be implicated in the recognition of object and spatial representations, respectively [145]. This is in line with work demonstrating distinct synaptic responses of the hippocampal dentate gyrus to entorhinal subdivision inputs [146]. In response to lateral entorhinal afferents, adult-born granule cells inhibit mature granule cells through group II metabotropic glutamate receptors to shape contextual representations. In response to medial entorhinal afferents, adult-born granule cells excite mature granule cells through N-methyl-D-aspartate receptors to shape spatial representations. Accordingly, questions remain on how the distinct entorhinal subdivisions relate to hippocampal-dependent emotional regulation as well as cognitive performance. Additionally, the hippocampal substructures related to emotional valences remain unresolved. For example, the temporal hippocampal region responsible for emotional processing [134] can also be further investigated in terms of the causal relationship between entorhinal stimulation and hippocampal neurogenesis that results in anti-depressive behaviors. Thus, the mechanism by which the entorhinal cortex is implicated in the regulation of hippocampal neurogenesis and hippocampal-dependent emotional and cognitive functions can be further scrutinized regarding the multifarious aspects of structural, neurophysiological, and electrophysiological interrelationships between the entorhinal cortex and hippocampus.

Secondly, future research is also required to elucidate the complex upper hippocampal circuitries that encompass diverse brain regions implicated in depression pathophysiology while maintaining the perspective of hippocampal neurogenesis. Novel evidence suggests that brain regions beside the entorhinal cortex are also related to dentate gyrus neurogenesis, suggesting that complex neural circuitries are implicated in hippocampal-dependent functions as well [147]. For example, deep brain stimulation of the ventromedial prefrontal cortex with high frequency leads to both upregulated neurogenesis-associated genes and enhanced hippocampal neuron proliferation. The prefrontal–hippocampal circuitry is linked to improvement in hippocampal-dependent object recognition [148]. Likewise, emotional memory circuitry is also proposed; basolateral amygdala activity controls hippocampal neurogenesis, fear context-specific proliferation, and recruitment of newborn neurons [149]. Anteromedial thalamic stimulation induces a 76% increase in the proliferation of progenitor neural stem cells in the hippocampal dentate gyrus [150]. Nonetheless, conflicting findings have shown that the prefrontal cortex and nucleus accumbens do not promote hippocampal neurogenesis [151]. On another level, complex hippocampal circuitry regulation is also marked by multiple signaling neurotransmitters, including serotonin from the dorsal and median raphe nuclei, acetylcholine from the septal nucleus and diagonal band of Broca, and dopamine from the ventral tegmental area [14]. The diverse neurotransmission signaling mechanisms also remain to be explored in terms of depression-related behaviors. Altogether, more efforts are required to elucidate how distinct brain regions individually and collectively contribute to neural circuitry-regulated neurogenesis that affects hippocampal-dependent cognitive and emotional functions.

## 7. Conclusions

As shown in Table 1, the causal relationship between entorhinal–hippocampal circuitry and neurogenesis and depression-related phenotype has been elucidated. The preclinical and clinical studies support the idea that the upper hippocampal circuitry may engage the entorhinal cortex in adult neurogenesis, which in turn drives a neural circuitry–neurogenesis model that can be a plausible concept to combat defects in memory, pattern separation, and mood, all of which are implicated in depression. In combating depression, the hippocampal neural stem cell niche is still regarded as a key functional target of anti-depressants and brain stimulation approaches to promote proliferation and maturation of dentate gyrus neurons. Regarding the linking of hippocampal neurogenesis to anti-depressive effects, entorhinal–hippocampal circuitry can be acquainted knowledge but also a novel area of depression pathophysiology, with a particular focus on the potential of the entorhinal cortex for regulating hippocampal neurogenesis, resulting in the enhancement of memory, pattern separation, and anti-depressive behaviors. Thus, entorhinal–hippocampal circuitry-regulated neurogenesis can be a plausible example of the neural circuitry–neurogenesis coupling model that enables further understanding of depression pathophysiology implicated in hippocampal-dependent cognitive and emotional symptoms in people with depression, and ultimately aids in the development of a more advanced therapeutic approach.

## Figures and Tables

**Figure 1 ijms-22-02468-f001:**
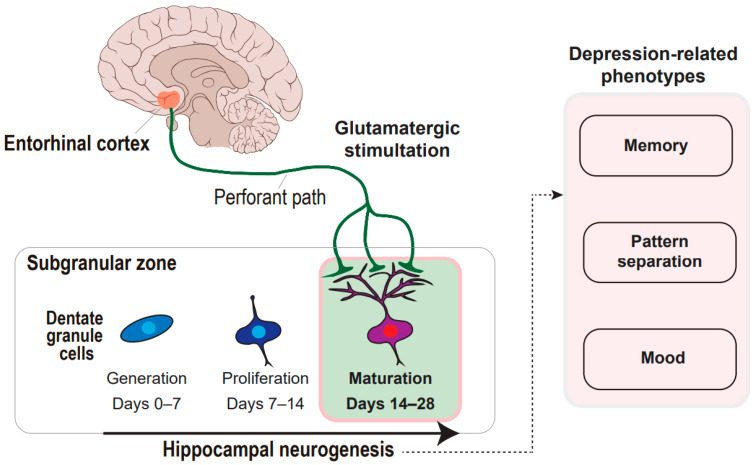
Concept framework of entorhinal cortex-regulated hippocampal neurogenesis in the regulation of depression-related phenotypes. The glutamatergic stimulation from the entorhinal cortex through the perforant paths to the subgranular zone of the hippocampal dentate gyrus is deciphered. The glutamatergic stimulation prompts the maturation of the dentate granule cells during hippocampal neurogenesis, which can regulate memory, pattern separation, and mood.

**Figure 2 ijms-22-02468-f002:**
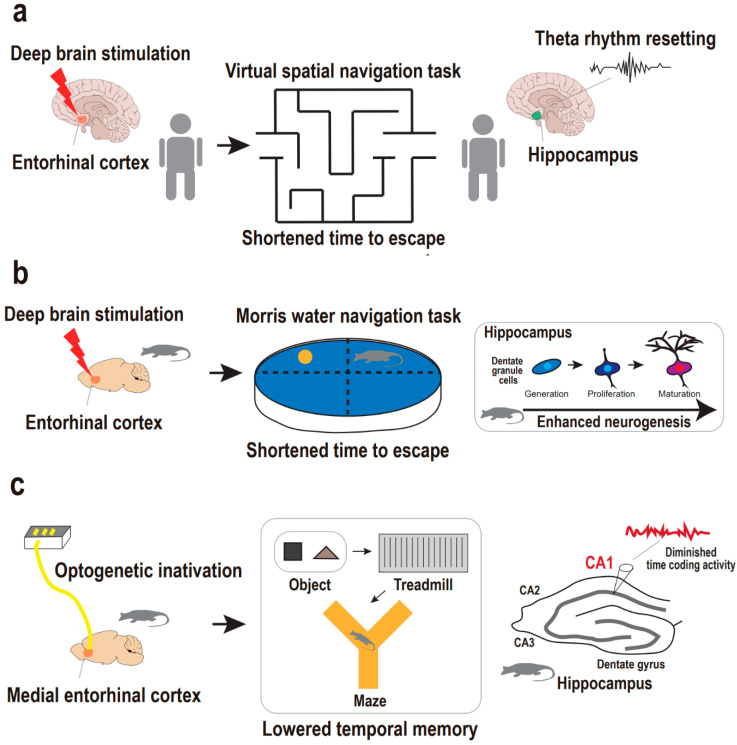
Supportive findings for entorhinal cortex-regulated hippocampal neurogenesis in the regulation of memory. (**a**) Entorhinal–hippocampal circuitry and hippocampal theta rhythm resetting in the regulation of human spatial memory. Deep brain stimulation on entorhinal cortex results in hippocampal theta rhythm resetting, which is accompanied with a shortened time to escape from a maze in the spatial navigation task in humans. (**b**) Entorhinal–hippocampal circuitry and neurogenesis in the regulation of animal spatial memory. Deep brain stimulation on entorhinal cortex results in enhanced neurogenesis, which is accompanied with a shortened time to escape in the Morris water navigation task in mice. (**c**) Entorhinal–hippocampal circuitry and hippocampal CA1 coding in the regulation of animal temporal memory. Optogenetic inactivation of the medial entorhinal cortex results in disruption in hippocampal CA1 coding activity, which is accompanied by the diminished temporal memory in the sequential object–treadmill–maze task in rats.

**Figure 3 ijms-22-02468-f003:**
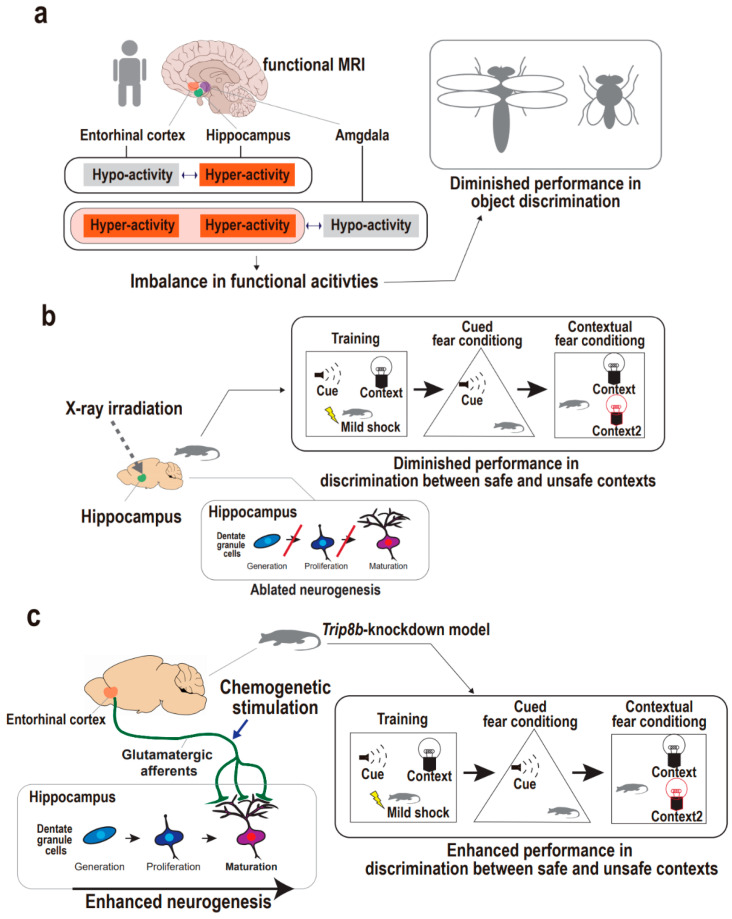
Supportive findings for entorhinal cortex-regulated hippocampal neurogenesis in the regulation of pattern separation. (**a**) Functional activities of upper hippocampal circuit regions including the entorhinal cortex in the regulation of human pattern separation. Imbalances in functional activities of the entorhinal cortex, hippocampus, and amygdala are accompanied with the diminished ability for object discrimination in humans. (**b**) Hippocampal neurogenesis in the regulation of animal pattern separation. X-ray irradiation on hippocampus results in ablated neurogenesis in mice, which demonstrate the diminished discrimination between the safe and unsafe representations in the contextual fear conditioning task. (**c**) Entorhinal–hippocampal circuitry and hippocampal neurogenesis in the regulation of animal pattern separation. Chemogenetic stimulation of entorhinal cortex results in enhanced neurogenesis in *Trip8b*-knockdown mouse, which is accompanied by the enhanced discrimination between safe and unsafe contexts in the contextual fear conditioning task.

**Figure 4 ijms-22-02468-f004:**
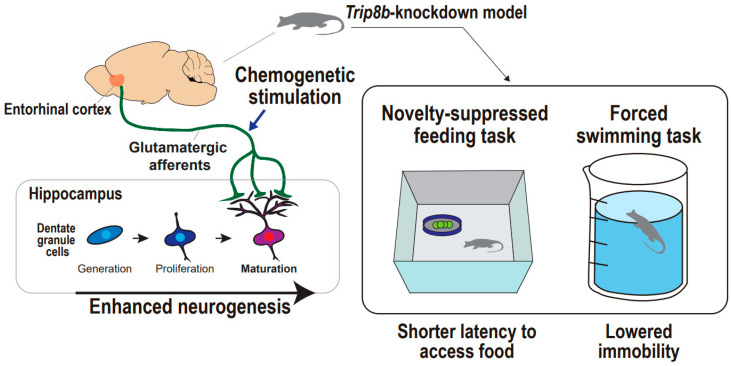
Supportive findings for entorhinal cortex-regulated hippocampal neurogenesis in the regulation of mood. Entorhinal–hippocampal circuitry and hippocampal neurogenesis in the regulation of animal mood. Chemogenetic stimulation of entorhinal cortex results in enhanced neurogenesis in *Trip8b*-knockdown mouse, which is accompanied by both shorter latency to access food in the novelty-suppressed feeding task and lowered immobility in the forced swimming task.

**Table 1 ijms-22-02468-t001:** Representative works elucidating the casual relationship between entorhinal–hippocampal circuitry and neurogenesis and depression-related phenotypes.

Stimulation Approach	Subject	Depression-Related Phenotype	Target of Stimulation	Consequence in Hippocampus	Behavioral Task	Work *
DBS	Human	Spatial memory	Entorhinal cortex	Hippocampal theta rhythm, resetted	Spatial navigation test	13
DBS	Mouse	Spatial memory	Entorhinal cortex	Dentate gyrus neurogenesis, enhanced	Morris water maze test	12
DBS	Rat	Spatial memory	Entorhinal cortex	Dentate gyrus neurogenesis, enhanced	Morris water maze test	15
Optogenetics	Transgenic rat	Temporal memory	Medial entorhinal cortex	Hippocampal CA1 temporal coding activity, enhanced	Object–treadmill–maze test	48
Chemogenetics	Transgenic mouse	Pattern separation	Entorhinal glutamatergic afferents	Dentate gyrus neurogenesis, enhanced	Fear-context conditioning test	16
Chemogenetics	Transgenic mouse	Depressive-like behaviors	Entorhinal glutamatergic afferents	Dentate gyrus neurogenesis, enhanced	Forced swimming test Novelty-suppressed feeding test	16

* Reference number in this article.

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
