# Peer review of "Neural Circuitry–Neurogenesis Coupling Model of Depression"

_ijms, 2021, doi:10.3390/ijms22052468_

Round 1
Reviewer 1 Report
The main topic of the reviewed manuxcript is a review of the latest scientific reports on the role of the entorhinal cortex in adult neurogenesis which supports the concept of the neural circuitry-neurogenesis and emotional and cognitive symptoms of depression. The individual chapters of the mauscript specifically describe the relationship between entorhinal-hippocampal circuitry and adult neurogenesis in memory, pattern separation, and mood. In conclusion, the authors propose suggestions for advancing the neural circuitry-neurogenesis coupling model of depression. Although the work is very interesting, a few gaps are noted. The most important is the lack of any collective form of the research results described (tables, charts, graphs), which would certainly facilitate the analysis of the described literature reports. These types of statements are very very important and valuable in reviews. Moreover, the authors use very elaborate sentences, which makes it difficult to understand the context. There is also a reservation about Conclusions. Already in the first sentence, the authors conclude that the evidence is weak, which in advance reduces the substantive value of the article. One should avoid expressing that something we have been working on turns out to be weak. The most important thing to improve is making and attachment of some graphic forms presenting/summarizing the scientific reports selected by the authors confirming their assumptions.
Author Response
Although the work is very interesting, a few gaps are noted. The most important is the lack of any collective form of the research results described (tables, charts, graphs), which would certainly facilitate the analysis of the described literature reports.
- We attached a figurative depiction and a table to the manuscript. Please refer to the figure 1 and table1, respectively.
Moreover, the authors use very elaborate sentences, which makes it difficult to understand the context.
There is also a reservation about Conclusions. Already in the first sentence, the authors conclude that the evidence is weak, which in advance reduces the substantive value of the article. One should avoid expressing that something we have been working on turns out to be weak.
- According to your advice, we removed the expression that might be negatively affecting the readership.
The most important thing to improve is making and attachment of some graphic forms presenting/summarizing the scientific reports selected by the authors confirming their assumptions.
- We attached a figure and a table that visualizes our concept frame and summarizes supportive findings for the concept, respectively.
Reviewer 2 Report
The manuscript entitled “Neural Circuitry-Neurogenesis Coupling Model of Depression” is interesting review. The association between neurogenesis and depression is important to investigate novel treatment of depression.
I have several comments.
It is sometimes difficult for readers to understand the contents, because the order of previous studies in humans and in animals is mixed. So, for readers, the authors should describe in a consistent order for each section. For instance, section 4, the authors described the studies in mice, then they described the studies in humans, and then they described studies in mice.
The authors emphasized deep brain stimulations (DBS), however, DBS is not used in the treatment of depression in general. Except treatment of antidepressants, Electro-convulsive therapy (ECT) is generally used. The authors should describe the associations between ECT and neurogenesis, if any.
Author Response
It is sometimes difficult for readers to understand the contents, because the order of previous studies in humans and in animals is mixed. So, for readers, the authors should describe in a consistent order for each section. For instance, section 4, the authors described the studies in mice, then they described the studies in humans, and then they described studies in mice.
- We revised our manuscript that can be better readable by separating between the human and animal findings for each of the sessions.
The authors emphasized deep brain stimulations (DBS), however, DBS is not used in the treatment of depression in general. Except treatment of antidepressants, Electro-convulsive therapy (ECT) is generally used. The authors should describe the associations between ECT and neurogenesis, if any.
- We appreciate your kind comments, and according to your valuable advice, diligently reviewed research papers that support the potential impact of ECT on neurogenesis1-7. After the thorough review, we again fully agreed to the importance of the ECT-provoked neurogenesis, but cannot find papers that support the relationship between entorhinal cortex and the ECT-provoked neurogenesis. As our scope of the manuscript underlies on the entorhinal circuitry-regulated hippocampal neurogenesis, we regret to say that the findings from ECT are not fully line with our points.
Additionally, ECT targets a whole brain rather than a specific region. Though there is a research that points to that structural neuroplasticity across neocortocal, limbic and paralimbic areas including entorhinal cortex can be promoted by ECT, this does elucidate neither the hippocampal neurogenesis nor the casual relationship between the increased thickness of entorhinal cortex and hippocampal changes8
Reviewer 3 Report
Dear Authors,
Neurogenesis is dependent on a neural circuitry activity. Defects in hippocampal neurogenesis were observed in individuals who suffered from depression. However, little is known how the neural circuitry defects impair neurogenesis in depression. The authors reviewed current findings of neural circuitry-regulated neurogenesis focusing on the entorhinal cortex regulated hippocampal neurogenesis, presenting evidence that the upper hippocampal circuitry engages the entorhinal cortex in memory, pattern separation, and mood regulation and proposing importance of neural circuitry-neurogenesis coupling studies in depression.
Please reconsider the following parts:
- Pages 1, Lines 27-40: Please rephrase the sentence to present incidence, prevalence, challenge, and the rationale of this manuscript in depression research.
- Pages 1, 2, Introduction: Please rephrase the sentences in a way that animal models support clinical findings in depression.
- Page 2, Lines 69-70: Please rephrase it.
- Pages, 2,3, Lines 63-102: Please present a figure depicting the entorhinal-hippocampal circuitry.
- Page 3,4, Section 3: Please separate human studies and animal studies clearly. Please present a table or figure summarizing the data presented.
- Page 4-6, Section 4: Please present a table or figure summarizing the data presented.
- Page 6,7, Section 5: Please present a table or figure summarizing the data presented.
- Page 8,9, Section 6: Please present a table or figure summarizing the data presented.
The manuscript contains no figure, no table, and 92 references. Please rephrase sentences in which the authors intend to present rationales throughout the manuscript. It deserves to clarify different methodologies such as human, animal, and clinical studies, to present them in figures and/or tables, and to describe their limitations. I recommend presenting data on large-scale brain network in the manuscript. Suggested reference: Balogh, L.; Tanaka, M.; Török, N.; Vécsei, L.; Taguchi, S. Crosstalk between Existential Phenomenological Psychotherapy and Neuroscience in Mood and Anxiety Disorders. Preprints 2020, 2020120625 (doi: 10.20944/preprints202012.0625.v1). I recommend including at least 150 references for review article. Citation and reference style need to be corrected. The manuscript carries great value presenting data on the entorhinal cortex-hippocampal circuit in memory, pattern recognition, and mood regulation and proposing potential benefit of neural circuitry-neurogenesis coupling studies in depression. I reconsider this manuscript for publication after major revision.
I declare no conflict of interest regarding this manuscript.
Best regards,
Author Response
(Reviewer3)
- Pages 1, Lines 27-40: Please rephrase the sentence to present incidence, prevalence, challenge, and the rationale of this manuscript in depression research.
- According to your kind comments, we added the informative epidemiologic findings of depression to the introduction of the manuscript. Also, we emphasized the challenges in current psychiatric intervention limitations and rationale for this review works that can help a better clinical approach to the refractory depression.
- Pages 1, 2, Introduction: Please rephrase the sentences in a way that animal models support clinical findings in depression.
- We rephrased the sentences in a way that readers can understand the study findings in animal models and human subjects separately.
- Page 2, Lines 69-70: Please rephrase it.
- Please refer to the rephrased sentence: “we review the neurophysiological findings that support the entorhinal cortex and hippocampus collectively contribute to adult neurogenesis.” (Previous sentence: “we address the neurophysiological findings linking the entorhinal cortex and hippocampus regarding adult neurogenesis.”)
- Pages, 2,3, Lines 63-102: Please present a figure depicting the entorhinal-hippocampal circuitry
- We made a figure that visualizes our concept frame of neural circuitry-neurogenesis coupling model in regulation of depression-related phenotypes.
- Page 3,4, Section 3: Please separate human studies and animal studies clearly. Please present a table or figure summarizing the data presented.
- We separated between human and animal studies clearly, and presented that with subheadings across the manuscript.
- We also made a figure and a table that can provide readers with a summary of our concept and supportive findings for that.
- Page 4-6, Section 4: Please present a table or figure summarizing the data presented.
- Page 6,7, Section 5: Please present a table or figure summarizing the data presented.
- Page 8,9, Section 6: Please present a table or figure summarizing the data presented.
- We provide the summarized figure and table that can be glimpsed at once for a better readership.
The manuscript contains no figure, no table, and 92 references.
- We made a figure and table that can be checked in our revised manuscript.
- We increased 92 to 150 references according to your kind advice.
Please rephrase sentences in which the authors intend to present rationales throughout the manuscript.
- We rephrased sentences through the manuscript in order to present clear rationales of our work. Please refer to the revised manuscript.
It deserves to clarify different methodologies such as human, animal, and clinical studies, to present them in figures and/or tables, and to describe their limitations.
- For a better comprehension of our works, we separated between human and animal studies, and also presented them in a descriptive figure and a summary table.
I recommend presenting data on large-scale brain network in the manuscript. Suggested reference: Balogh, L.; Tanaka, M.; Török, N.; Vécsei, L.; Taguchi, S. Crosstalk between Existential Phenomenological Psychotherapy and Neuroscience in Mood and Anxiety Disorders. Preprints 2020, 2020120625 (doi: 10.20944/preprints202012.0625.v1).
- We deeply appreciate a clear guidance for the construction of a new figure for our manuscript. We carefully scrutinized the reference article you recommended, and according to your recommend example, made a figurative depiction of our main theme: Neural circuitry-regulated neurogenesis and depression-related phenotypes. Please refer to the new figure in our revised manuscript.
I recommend including at least 150 references for review article. Citation and reference style need to be corrected.
References
- Madsen, T.M. et al. Increased neurogenesis in a model of electroconvulsive therapy. Biological psychiatry 47, 1043-1049 (2000).
- Hellsten, J. et al. Electroconvulsive seizures increase hippocampal neurogenesis after chronic corticosterone treatment. European Journal of Neuroscience 16, 283-290 (2002).
- Wennström, M., Hellsten, J. & Tingström, A. Electroconvulsive seizures induce proliferation of NG2-expressing glial cells in adult rat amygdala. Biological psychiatry 55, 464-471 (2004).
- Madsen, T.M., Yeh, D.D., Valentine, G.W. & Duman, R.S. Electroconvulsive seizure treatment increases cell proliferation in rat frontal cortex. Neuropsychopharmacology 30, 27-34 (2005).
- Perera, T.D. et al. Antidepressant-induced neurogenesis in the hippocampus of adult nonhuman primates. Journal of Neuroscience 27, 4894-4901 (2007).
- Wennström, M., Hellsten, J., Ekstrand, J., Lindgren, H. & Tingström, A. Corticosterone-induced inhibition of gliogenesis in rat hippocampus is counteracted by electroconvulsive seizures. Biological psychiatry 59, 178-186 (2006).
- Jansson, L., Wennström, M., Johanson, A. & Tingström, A. Glial cell activation in response to electroconvulsive seizures. Progress in Neuro-Psychopharmacology and Biological Psychiatry 33, 1119-1128 (2009).
- Pirnia, T. et al. Electroconvulsive therapy and structural neuroplasticity in neocortical, limbic and paralimbic cortex. Translational psychiatry 6, e832-e832 (2016).
Round 2
Reviewer 1 Report
The authors took into account all the suggestions, so I accept the manuscript in present form.
Author Response
I greatly appreciate your kind comments.
Reviewer 2 Report
In this revised manuscript entitled “Neural Circuitry-Neurogenesis Coupling Model of Depression”, Kim et al have adequately addressed the reviewer’s comments. The manuscript is now significantly improved. Therefore, I only recommended a minor suggestion.
If possible, the authors should make brief table or figure of summary like Figure 1 or Table 1 per section, and it can be easy for readers to understand the contents of this review.
Author Response
We greatly appreciate your kind comments. We have complemented the tables.
Reviewer 3 Report
Dear Authors,
Neurogenesis is dependent on a neural circuitry activity. Defects in hippocampal neurogenesis were observed in individuals who suffered from depression. However, little is known how the neural circuitry defects impair neurogenesis in depression. The authors reviewed current findings of neural circuitry-regulated neurogenesis focusing on the entorhinal cortex regulated hippocampal neurogenesis, presenting evidence that the upper hippocampal circuitry engages the entorhinal cortex in memory, pattern separation, and mood regulation and proposing importance of neural circuitry-neurogenesis coupling studies in depression.
Please reconsider the following parts:
- Pages 7-10, “memory”: Please present a table or figure summarizing the data of human studies.
- Pages 10-14, “Pattern separation”: Please present a table or figure summarizing the data of human and animal studies.
- Pages 14-7, “Mood regulation”: Please present a table or figure summarizing the data presented.
The manuscript contains one figure, one table, and 151 references. Many phrases were revised; one figure and one table were added; and the number of references was expanded in the manuscript. However, there is no table or figure for human studies for memory, pattern separation, and mood regulation. The manuscript carries great value presenting data on the entorhinal cortex-hippocampal circuit in memory, pattern recognition, and mood regulation and proposing potential benefit of neural circuitry-neurogenesis coupling studies in depression. I reconsider this manuscript for publication after major revision.
I declare no conflict of interest regarding this manuscript.
Best regards,
Author Response
We greatly appreciate your kind comments. We have complemented the table according to your comments.
Round 3
Reviewer 3 Report
13 February 2021
Review on the manuscript titled “Neural Circuitry-Neurogenesis Coupling Model of Depression” by Kim IB et al, submitted to International Journal of Molecular Sciences.
Dear Authors,
Neurogenesis is dependent on a neural circuitry activity. Defects in hippocampal neurogenesis were observed in individuals who suffered from depression. However, little is known how the neural circuitry defects impair neurogenesis in depression. The authors reviewed current findings of neural circuitry-regulated neurogenesis focusing on the entorhinal cortex regulated hippocampal neurogenesis, presenting evidence that the upper hippocampal circuitry engages the entorhinal cortex in memory, pattern separation, and mood regulation and proposing importance of neural circuitry-neurogenesis coupling studies in depression.
The manuscript contains four figures, one table, and 151 references. More figures were added, which help readers understand the manuscript and thus have made it more informative. The manuscript carries great value presenting data on the entorhinal cortex-hippocampal circuit in memory, pattern recognition, and mood regulation and proposing potential benefit of neural circuitry-neurogenesis coupling studies in depression. I accept this manuscript for publication in current form.
I declare no conflict of interest regarding this manuscript.
Best regards,
Masaru Tanaka, M.D., Ph.D.
Author Response
 According to the reviewer 2’s valuable advice, we generated additional three summary figures for the sections that suggest entorhinal cortex-regulated hippocampal neurogenesis and depression-related phenotypes including disturbances in memory, pattern separation, and mood.